# New Look at RSV Infection: Tissue Clearing and 3D Imaging of the Entire Mouse Lung at Cellular Resolution

**DOI:** 10.3390/v13020201

**Published:** 2021-01-28

**Authors:** Maxence Frétaud, Delphyne Descamps, Daphné Laubreton, Marie-Anne Rameix-Welti, Jean-François Eléouët, Thibaut Larcher, Marie Galloux, Christelle Langevin

**Affiliations:** 1Université Paris-Saclay, INRAE, UVSQ, VIM, 78350 Jouy-en-Josas, France; maxence.fretaud@inrae.fr (M.F.); delphyne.descamps@inrae.fr (D.D.); daphne.laubreton@inserm.fr (D.L.); jean-francois.eleouet@inrae.fr (J.-F.E.); 2CIRI, Centre International de Recherche en Infectiologie, Univ Lyon, Inserm, U1111, Université Claude Bernard Lyon 1, CNRS, UMR5308, ENS de Lyon, F-69007 Lyon, France; 3Université Paris-Saclay, INSERM, Université de Versailles St. Quentin, UMR 1173 (2I), 78000 Versailles, France; marie-anne.welti@aphp.fr; 4Assistance Publique Hôpitaux de Paris, Université Paris Saclay, Hôpital Ambroise Paré, Laboratoire de Microbiologie, 92100 Boulogne-Billancourt, France; 5INRAE, Oniris, UMR 703 APEX, 44307 Nantes, France; thibaut.larcher@inrae.fr; 6INRAE, IERP, Université Paris-Saclay, 78350 Jouy-en-Josas, France

**Keywords:** RSV infection, tissue clearing, 3D imaging of lungs, inclusion bodies, viral pathophysiology, RSV tropism

## Abstract

Background: Respiratory Syncytial Virus (RSV) is the major cause of severe acute respiratory tract illness in young children worldwide and a main pathogen for the elderly and immune-compromised people. In the absence of vaccines or effective treatments, a better characterization of the pathogenesis of RSV infection is required. To date, the pathophysiology of the disease and its diagnosis has mostly relied on chest X-ray and genome detection in nasopharyngeal swabs. The development of new imaging approaches is instrumental to further the description of RSV spread, virus–host interactions and related acute respiratory disease, at the level of the entire lung. Methods: By combining tissue clearing, 3D microscopy and image processing, we developed a novel visualization tool of RSV infection in undissected mouse lungs. Results: Whole tissue analysis allowed the identification of infected cell subtypes, based on both morphological traits and position within the cellular network. Furthermore, 3D imaging was also valuable to detect the cytoplasmic viral factories, also called inclusion bodies, a hallmark of RSV infection. Conclusions: Whole lung clearing and 3D deep imaging represents an unprecedented visualization method of infected lungs to allow insight into RSV pathophysiology and improve the 2D histology analyses.

## 1. Introduction

The respiratory syncytial virus (RSV) belongs to the *Mononegavirales* order and the *Pneumoviridae* family [1]. It is the most common cause of severe respiratory tract disease in young children worldwide and is responsible for millions of hospitalizations and thousands of deaths per year [2,3]. RSV is also recognized as a major pathogen of elderly and immunocompromised patients who often develop severe lower airway complications, whereas the infection is generally restricted to upper airways and associated with symptoms of mild cold in healthy adults [2,4]. Despite its huge impact in human health, there is still no vaccine against RSV. The humanized monoclonal antibody (Palvizumab) constitutes the only specific treatment available used for passive prophylaxis [5,6]. It is, however, restricted to high-risk pediatric infants because of moderate efficacy and high cost (>5000 € per treatment). Infection begins in the upper respiratory tract before spreading to the lower respiratory system (bronchia and alveoli), thereby accounting for severe pulmonary inflammation (bronchiolitis and pneumonia). The severity of the symptoms and the disease outcome depend on virus load, viral cytopathology and genetic polymorphism of the patients [7,8,9]. Moreover, host immune responses restrict the spread of the virus altering its tropism, which impedes the development of the disease. So far, viral tropism has been characterized by the identification of RSV infected cells in lung biopsies from RSV infected patients [10,11] and from established models of airways epithelium [12], such as recently developed three-dimensional (3D) cultures [13,14]. RSV has been shown to actively replicate in the lung ciliated epithelial cells, type I and type II alveolar pneumocytes, and can be detected in alveolar macrophages [15,16]. In addition, recent evidence demonstrated the capacity of RSV to replicate in neurons both in vitro in N2a neuronal cells [17,18] and in vivo in the olfactory neurons of the olfactory mucosa [19]. In these cases, infected cells have been identified by immunostaining of RSV antigens. Such immunostaining were also used to evaluate the formation of cytoplasmic inclusion bodies (IBs) [20,21] and RSV-induced syncytia (observed in post-mortem lung analysis of RSV infected patients) [15], which constitute hallmarks of RSV infection. Importantly, IBs are common features of *Mononegavirales* infections [22,23,24,25,26,27,28,29], used in histology for the diagnosis of rabies virus since their discovery in 1903 [30,31]. Initially considered as aggresomes formed by viral dead-end products, these membrane-less structures were recently shown to be viral factories in which both viral replication and transcription occur [21,32]. In addition, recent evidence has shown that they could also be involved in the modulation of innate immune responses [31,33,34].

Recent advances in tissue clearing techniques (which aims to render tissue optically transparent through reduction of light scattering and light absorption) have fostered the emergence of deep imaging methods of large samples without the need for physical sectioning. A large number of methods have been described, grouped into four main categories based on the chemistry used for the clearing. They all consist in homogenization of refractive index of the whole tissue achieved by the use of organic solvents, high refractive index aqueous solutions, hyperhydrating solutions or hydrogel embedding. Once transparent, the samples; i.e., biopsies, organoids or organisms, can be observed to describe complex biological networks in three dimensions at cellular resolution. Therefore, combination of tissue clearing and deep imaging opened unprecedent opportunities in term of anatomy, diagnosis or host-pathogen interaction studies. Clearing protocols developed on entire mouse brain quickly evolved into the treatment of other organs, organoids [35,36] and entire mouse [37]. More specifically, mice lung clearing has been used to describe lung morphology [38,39], alveolar development [40], bronchus-associated lymphoid tissue [41], together with the histopathology of pulmonary illness [42,43,44]. This approach was also used to investigate bacterial infections such as mycobacterium granuloma (organization and angiogenesis) [45,46] and *Aspergillus fumigatus* infection [47]. However, up to now, no viral infections have been characterized at the level of the whole lung by using 3D imaging. We recently explored the distribution of RSV infection cells in the nasal cavity of infected mouse, through development of tissue clearing and 3D light sheet imaging [19]. This allowed the observation of individual infected cells throughout the whole tissue [19,41].

Here, we pursued the characterization of the tropism of RSV in mouse lung at the level of the whole organ. Deep tissue imaging using light-sheet or 2-photon microscopy allowed us to visualize the distribution of the RSV infected cells in their tissue environment. 3D reconstruction of acquired images provided precise description of the morphology of the infected cells. This has also been essential for evaluating the relationship of the different cell subtypes and determine their spatial distribution with respect to the respiratory epithelium and of the alveoli wall. We also investigated RSV-induced pulmonary disease through observation of tissue integrity and potential obstruction of lumens of the bronchiolar airways. Finally, the approach permitted detection of the cytoplasmic viral IBs in the lung parenchyma. Overall, our results describe novel visualization tools opening the way to the study of pathogen co-infections, comparison of RSV tropism in different physiological conditions (ages, immunodepression or chronic inflammation) or host pathogen interaction.

## 2. Materials and Methods

### 2.1. Virus Production

Recombinant RSV viruses rHRSV-mCherry and rHRSV-Luc corresponding to RSV Long strain expressing either the mCherry or the Luciferase proteins respectively were propagated in HEp-2 cells (ATCC number CCL-23) by infecting a freshly prepared confluent monolayer grown in MEM supplemented with 2% of FCS as previously described [48]. When the cytopathic effect involved the whole monolayer, the viral suspension was collected from the infected cell supernatant and clarified by centrifugation. Titration of the rHRSV-mCherry and rHRSV-Luc virus stocks was determined by plaque assay on HEp-2 cell monolayers at 1.2 × 10^6^ pfu/mL and 1.4 × 10^7^ pfu/mL, respectively.

### 2.2. Ethic Statements

The in vivo work of is study was carried out in accordance with INRAE guidelines in compliance with European animal welfare regulation. The protocols were approved by the Animal Care and Use Committee at “Centre de Recherche de Jouy-en-Josas” (COMETHEA) under relevant institutional authorization (“Ministère de l’éducation nationale, de l’enseignement supérieur et de la recherche”), under authorization number 2015060414241349_v1 (APAFIS#600). All experimental procedures were performed in a Biosafety level 2 facilities.

### 2.3. Mice and Viral Infection

Adult BALB/c mice were purchased from Janvier (Le Genest, St. Isle, France) and housed under Bio-Safety Level-2 conditions (IERP, INRAE, Jouy-en-Josas, France). Adult (6–8 weeks-old) mice were anesthetized with a mixture of ketamine and xylazine (1 and 0.2 mg per mouse, respectively) and received 50 μL of rHRSV-mCherry (7 × 10^5^ pfu) or rHRSV-Luc (6 × 10^4^ pfu) or cell culture media as mock-infection control by intranasal (in) instillation [48].

### 2.4. Sample Collection

Mice were euthanized at different time points by intraperitoneal injection of pentobarbital (150 μL). Intracardiac perfusion with 10 mL PBS was performed before collecting the lungs, nasal turbinates (NT) or trachea which were directly frozen in nitrogen and conserved at −80 °C until being processed for quantification of luciferase activity. For tissue clearing, the lungs were fixed in 4% paraformaldehyde for 24 h, before transfer in 70% ethanol.

### 2.5. Bioluminescence Measurements

Photon emission was measured in the lungs and nose of rHRSV-Luc or Mock infected mice using the IVIS system (Xenogen Biosciences) and Living Image software (version 4.0, Caliper Life Sciences, Hopkinton, MA, USA). Adult mice received 50 μL of D-luciferin (30 mg/mL, Perkin Elmer, Waltham, MA, USA) by instillation and luciferase activity was measured for 1 min with f/stop = 1 and binning = 8. A digital false-color photon emission image of the mouse was generated, and photons were counted within a constant region of interest corresponding to the surface of the whole lungs or nose area. Results are expressed in radiance (photons/sec/cm^2^/sr). Photon emission was also analyzed in lungs, nasal turbinates or trachea homogenized in 300 μL of Passive Lysis Buffer (1 mM Tris pH 7.9; 1 mM MgCl_2_; 1% Triton × 100; 2% glycerol; 1 mM DTT) with a Precellys 24 bead grinder homogenizer, 2 × 15 s at 4 m/s, by mixing 100 µL of lysate with 100 µL of D-luciferin (0.1 mg/mL) supplemented with ATP (0.5 M) in flat bottom 96-well black plates (Thermo Scientific, Waltham, MA, USA. Luciferase activity was measured for 1 min with f/stop = 1 and binning = 8. Radiance was normalized to the weight of tissues.

### 2.6. Immunohistochemistry and Tissue Clearing

Immunostaining and clearing of whole mouse lungs was performed following the iDISCO+ protocol [49]. Alexa Fluor 488 Streptavidin (S11223, ThermoFisher Scientific), anti-mCherry rabbit IgG (600-401-P16, Rockland) and the secondary antibody Alexa Fluor 594-conjugated goat anti-rabbit IgG (A11012, ThermoFisher Scientific) were diluted at 1:500 in PBS + 10% DMSO + 0.2% triton X−100 + 100 µg/mL saponin + 10 µg/mL heparin + 3% horse serum + 0.05% sodium azide. Lungs were incubated 9 days at 37 °C successively with first and secondary antibodies.

Biopsies of mouse lungs were immunostained with anti RSV antibodies according to the iDISCO+ protocol [49]. The rabbit polyclonal anti-Nucleoprotein of RSV [50] and the secondary antibody Alexa Fluor 488-conjugated goat anti-rabbit IgG (A11034, ThermoFisher Scientific) were diluted at 1:500 in PBS + 10% DMSO + 0.2% triton X−100 + 100 µg/mL saponin + 10 µg/mL heparin + 3% horse serum + 0.05% sodium azide. Biopsies were incubated for 9 days at 37 °C successively with first and secondary antibodies. For counterstaining, biopsies were incubated successively in 20% DMSO/PBS, 70% DMSO/PBS and 100% DMSO during 15 min for each step. Membrane and cytoplasm were stained with the lipophilic dye DiI (D3911, ThermoFisher Scientific) diluted at 0.001% in DMSO overnight at room temperature. Counterstained biopsies were rinsed successively in 100% DMSO, 50% DMSO/PBS and three times in PBS. Nuclei were stained with DAPI (D9542, Sigma-Aldrich, Darmstadt, Germany) diluted at 0.5 µg/mL in PBS overnight at room temperature. Stained tissues were cleared with the Refractive Index Matching Solution described in Yang et al. [51]. RIMS was prepared by diluting HistoDenz (D2158, Sigma-Aldrich) at 1.33 g/mL in PBS + 0.05% sodium azide. Biopsies were incubated for 2 days at room temperature in RIMS before mounting and imaging.

### 2.7. 3D Imaging

Whole mouse lungs were acquired with a light-sheet ultramicroscope (Lavision Biotech, Bielefeld, Germany) using a 2X objective and a PCO Edge SCMOS CCD camera (2.560 × 2.160 pixel size, LaVision BioTec). Lungs biopsies were acquired with Leica SP8 confocal/two-photon microscope using a HCX IRAPO L 25X/0,95NA water immersion objective (Leica microsystems, Wetzlar, Germany). For identification of RSV infected cells, DAPI and Alexa Fluor 594 were both excited at 800 nm with a Chameleon Vision II laser (Coherent, Lisse, France) and fluorescence was detected by NDD HyD detectors (Leica microsystems) with BP 525/50 and BP585/40 filters respectively. For visualization of IBs, DAPI, Alexa-488 and DiI were excited with 405 nm, 488 nm and 552 nm lasers respectively.

### 2.8. Image Analysis

Images were processed with Fiji for cropping, brightness and contrast adjustments and denoising. Maximal projection of whole lung and 3D segmentation were generated with Imaris (Oxford Instruments, Abigdon, UK) with Section and Surfaces modules respectively. Counting of RSV foci was performed with the Spots module with an estimated diameter of 40 µm. The resulting spots were filtered on standard deviation value to remove false positive spots due to high background.

## 3. Results and Discussion

### 3.1. 3D Imaging of the Sites of Replication of RSV in Mice

Generation of recombinant human RSV expressing Luciferase (rHRSV-Luc) has been instrumental in following RSV infection processes (spreading and replication) in whole living mice using bioluminescent intravital imaging [48]. Replication of rHRSV-Luc in vivo in Balb/c mice was quantified by luminescence in the whole mouse body during the first 2 days post-infection (d.p.i.). The RSV signal was initially detected mainly in the nasal cavity (Figure 1A), before spreading to the lower respiratory tract, reaching a peak of virus replication in the lungs at 4 d.p.i. (Figure 1B), as previously published [48]. To overcome the potential limits of detection of photons scattered through thick tissues, we analyzed bioluminescence signals of rHRSV-Luc in specific tissues of the upper and lower airways, i.e., nasal turbinates, the trachea and the lungs. Quantification of Luciferase activity in tissue lysates showed that at 4 d.p.i., rHRSV-Luc is detected in the nasal turbinates and in the lungs but not in the trachea (Figure 1C). These results can be related to the dose of the virus administered by intranasal instillation and do not exclude the ability of RSV to infect tracheal cells in other conditions. Notably, it was previously shown that viral replication estimated in vivo by quantifying bioluminescence signals directly correlated with virus load [48].

To better characterize the spatial distribution of the RSV-infected cells in the lungs, we then performed tissue clearing and deep 3D imaging approaches, as previously reported [19]. To this end, mice were intranasally infected with recombinant human RSV expressing mCherry (rHRSV-mCherry) and sacrificed at 2 or 4 d.p.i. to collect and fix the lungs. The methanol-based dehydration step used in our clearing protocol is detrimental to mCherry fluorescence. Observation of mCherry required whole-mount immunolabeling using the I-DISCO^+^ protocol [49]. Co-staining with Streptavidin coupled to Alexa 488 was also used to label the airways as reported by Scott et al. [39]. Finally, immunolabeled lungs were cleared using the iDISCO^+^ protocol [49]. Indeed, clearing is essential for deep visualization of RSV infected cells, together with the entire bronchial tree in intact mouse lungs using light-sheet microscopy (Figure 2A). Sparse mCherry^+^ cells were detected at 2 d.p.i. distributed in the whole lung parenchyma. As previously observed with luminescence quantification of rHRSV-Luc infection (Figure 1A,B), the number of mCherry^+^ cells increased at 4 d.p.i. and the cells remained scattered throughout the whole tissue. The precise distribution of mCherry+ cells within the lungs was established by image processing (Figure 2B,C), enabling quantification of spots (mCherry^+^ cells) in a region of interest as shown in Figure 2B–D, where 1519 spots were detected in a portion of tissue of 59.9 mm^3^. Spots were classified according to their distance to the airway epithelium. Two groups were distinguished from a threshold established at 4 µm corresponding to the limit of resolution (pixel size). According to these parameters, the analyses showed that 23.3% of the spots were localized at a distance less than or equal to 4 µm of the respiratory tract (Figure 2D), and thus were considered to be part of the airway epithelium (Figure 2D). In contrast, the vast majority of the detected spots were distributed at a distance of more than 4 µm from the airways and scattered throughout the whole tissue, suggesting that RSV can spread deep in the pulmonary tissue (Figure 2C). In conclusion, the processing of 3D image data sets allowed the visualization of RSV infectious sites in undissected infected lungs at the cellular resolution. This provided quantitative analyses and spatial information of the RSV infected cells to complement bioluminescent intravital imaging of lower spatial resolution.

### 3.2. rHRSV-mCherry is Detected in Pneumocytes and Epithelial Cells

To further investigate the populations of infected cells in our model, biopsies of 4 d.p.i mouse lungs were immunolabeled with anti-mCherry and DAPI before clearing in the refractive index matching solution (RIMS) [51]. High resolution 2-photon imaging of biopsies (5 mm^3^) allowed the detection of mCherry in different cell subtypes identified based on their morphology, localization and relationship with other cells within the lung tissue (Figure 3A–D). A few isolated mCherry^+^ columnar to cuboidal cells were identified as epithelial cells of the respiratory epithelium of terminal respiratory airways (Figure 3A). mCherry, was also abundantly expressed in flattened cohesive cells with a plump nucleus lining alveolar walls (Figure 3B) and therefore identified as type I pneumocytes. A few scattered mCherry^+^ round cells with prominent nuclei were additionally identified (Figure 3C,D). 3D reconstruction of the image acquisitions presented in Figure 3E,F have been essential to identify and discriminate mCherry^+^ alveolar macrophages (Figure 3C,F) free in the alveolar lumen (Figure 3F) from the mCherry^+^ type II pneumocytes (Figure 3D,G) lining the alveoli wall (Figure 3G). Notably, fluorescence sporadically detected in macrophage cytoplasm could result from direct RSV gene expression or more probably to phagocytosis of RSV-infected cells.

No evidence of strong RSV-induced damage was detected in the lung biopsies (respiratory epithelium hyperplasia, necrosis, inflammatory cell infiltration, BALT hyperplasia) except a few desquamated cells of the respiratory epithelium in bronchial lumen (Figure 3E). Overall, the detection of RSV infected cells in the whole tissue gave insights in the RSV tropism which mainly target the type I pneumocytes. Although virus pathogenicity has been examined, infection did not elicit a strong inflammatory response in our model. These observations correlate with previous data showing that in this model RSV replication occurred from 2 to 4 d.p.i. before observation of immune cell infiltration from 5 to 7 d.p.i., leading to tissue damage [52,53]. Further studies may be considered to characterize more precisely the activation of resident or recruited immune cells in response to the viral infection, at different time points, and using specific immunolabeling of lymphocytes, leucocytes and/or macrophages. In addition, the tissue damages commonly triggered by respiratory viral infections (epithelium disruption, airway obstructions, leukocyte infiltration, etc.) may be investigated with these new techniques in order to improve the knowledge acquired from X-ray Chest images and histological analysis.

### 3.3. Deep-Imaging of Clarified Mouse Lungs Allows the Visualization of RSV Inclusion Bodies in the Whole Tissue

We next assessed the detection of the cytoplasmic IBs using our approach. These viral factories [21] contained all the constituents of the RSV polymerase complex, composed of four proteins (i.e., the RNA-dependent RNA polymerase L, its cofactor P, the ribonucleocapside composed of viral genome encapsidated by the N protein, and the transcription factor M2-1) [4]. IBs cannot be detected in biopsies when immunolabeled with anti-mCherry (Figure 3A–D), because in the rHRSV-mCherry, mCherry is expressed as an additional protein (i.e., not fused to a viral protein) with a cytosolic expression pattern [48]. We thus performed immunostaining of lung biopsies with anti-N antibodies together with counterstaining of nuclei with DAPI and cell membrane and cytoplasm with DiI (Figure 4A–C). Deep-imaging revealed the presence of N-positive fluorescent inclusions scattered throughout the lung parenchyma (Figure 4A–C). Higher magnification of positive signals confirmed the presence of cytosolic fluorescent puncta in type I pneumocytes (Figure 4C). Notably, similar results were obtained when labeling the P protein (Figure 4D,E). These observations strongly suggest that cytosolic puncta could correspond to RSV IBs, although a co-staining by both anti-N and anti-P antibodies is needed to confirm this hypothesis [20,21]. These last observations show that 3D imaging of transparent lung biopsies is likely to constitute a valuable tool for the assessment of RSV infection at a subcellular level.

## 4. Conclusions

Obtaining and processing 3D image datasets opens new insights into the understanding of RSV infection in complement to 3D transmission electron microscopy, electron tomography and focused ion beam scanning electron microscopy, to which it should be compared (for review; [54]). Indeed, 3D visualization of RSV infection in large tissues at subcellular resolution will be instrumental in describing the distribution of the RSV and other respiratory pathogens along the airway or sparse in the parenchyma, the activated leukocytes and the tissue damage. Mice are often used as models for RSV infection studies. However, the relevance of the mouse model for the study of RSV infection is still debated, as mice do not completely recapitulate the RSV human disease. However, the established method will be suitable to support basic research on viral infections of lungs from various species [55]. This will enable the enrichment of the knowledge acquired through intravital imaging (bioluminescence) [19,48], medical X-ray imaging [56] and classic 2D histology from micrometric sections of tissue [19]. The use of tissue clearing in human health—and more particularly for respiratory diseases—emerges and has already been reported for the analysis of lung organoids or lung biopsies from human patients [39,57]. In combination with biochemistry and histology, tissue clearing and 3D imaging will contribute to personalized medicine [58].

Furthermore, 3D reconstruction of the vascular, neuronal or airway networks in undissected lungs could also trigger the discovery of discrete and/or complex physiopathologies associated with RSV infection. Finally, these approaches open the way for the study of the RSV infection in different tissue environments with respect to its physiological contexts (newborns/adults—immunocompromised or auto-inflammatory status).

## Figures and Tables

**Figure 1 viruses-13-00201-f001:**
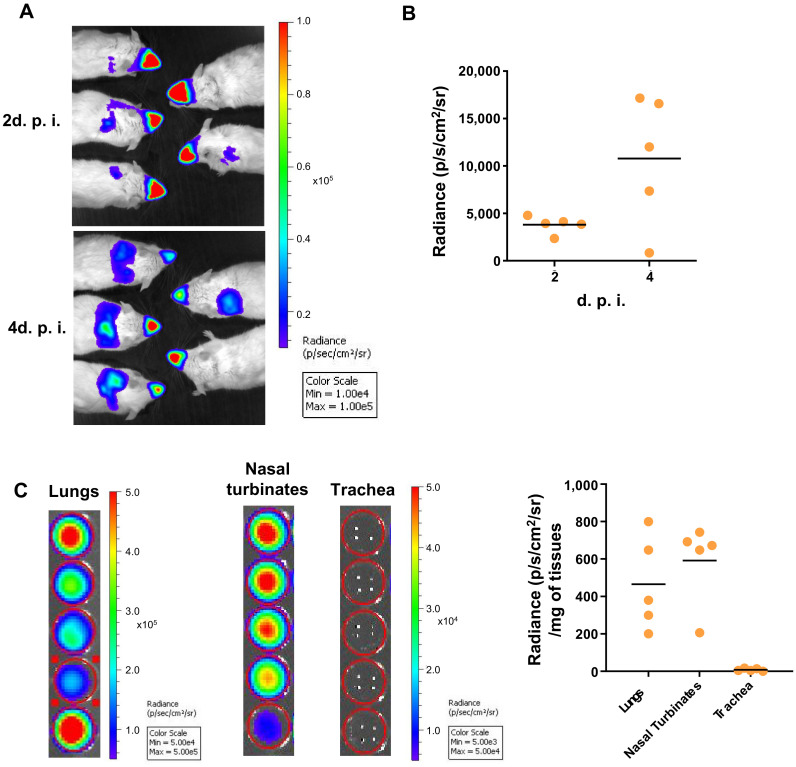
RSV replication in whole BALB/c mice. Adult BALB/c mice were intranasally infected with rHRSV-Luc. (**A**) At 2- and 4-days post-infection (d.p.i.), mice intranasally received luciferin (1.5 mg under 50 µL), and luciferase activity was visualized in the IVIS-200 imaging system. Images make it possible to observe virus replication in the nose and the lungs of animals at 2- and 4- days post-infection. The scale on the right indicates the average radiance: the sum of the photons per second from each pixel inside the region of interest/number of pixels (p/s/cm2/sr). A red signal indicates a high radiance, and consequently a high luciferase activity, whereas a blue signal indicates a low radiance, indicative of a low replication. (**B**) Quantification of luciferase activity in the lungs of living mice (radiance in photon/s/cm^2^/sr). (**C**) Quantification of luciferase activity in lungs, nasal turbinates (NT) or trachea homogenates. The scales indicate the average radiance: the sum of the photons per second from each pixel inside the region of interest/number of pixels (p/s/cm2/sr). Individual results were expressed radiance in photon/s/cm^2^/sr/mg of tissues and means are expressed as mean of individual ± SEM (one representative experiment of two with *n*= 5 to 8 mice/group) are represented (*n*= 5 mice/group).

**Figure 2 viruses-13-00201-f002:**
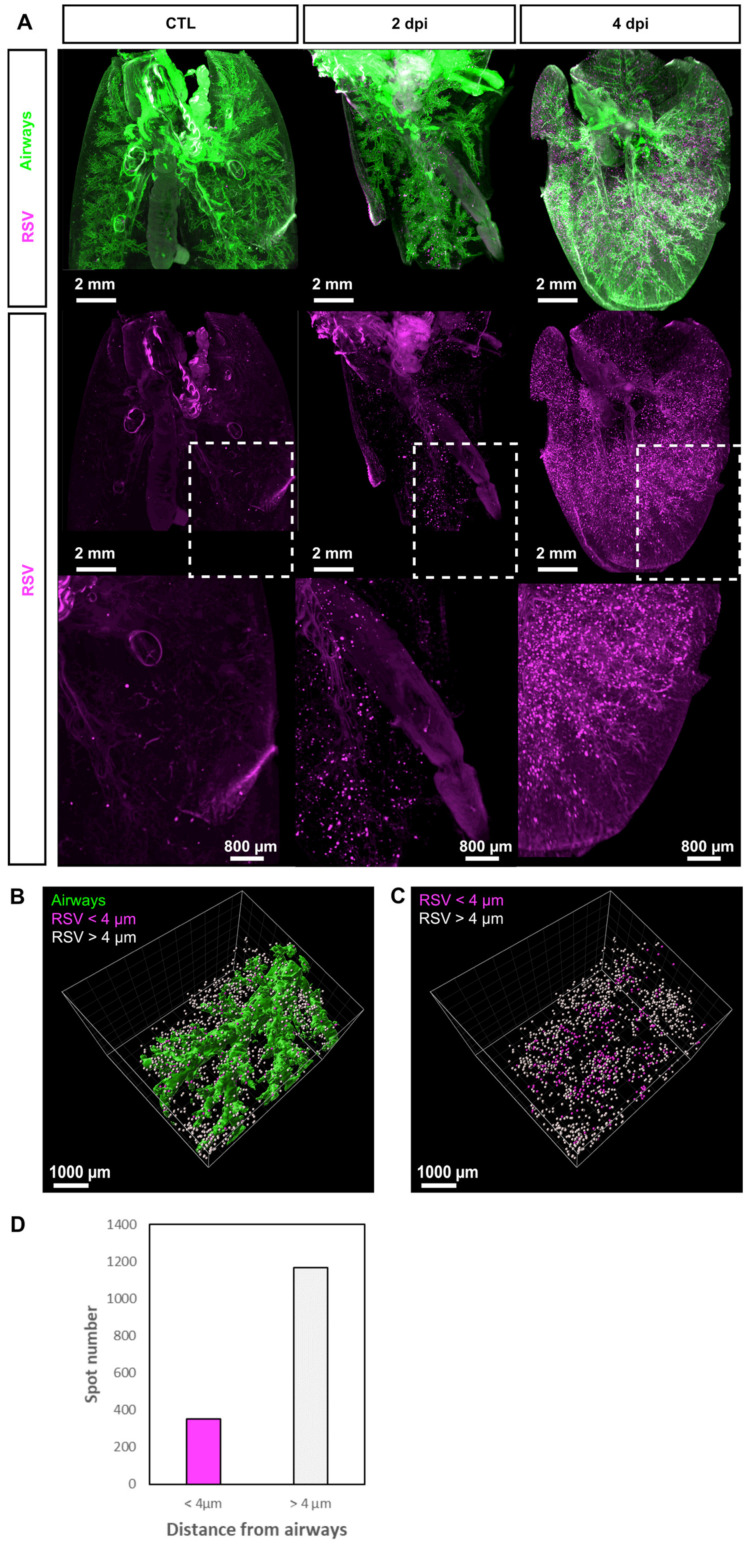
Distribution of RSV infected cells in whole mouse lung. (**A**) Maximal projections of light sheet acquisitions from whole mouse lungs sampled at 2 and 4 d.p.i. Lungs from mock-infected mouse were considered as control. In toto immunolabeling was processed with Streptavidin coupled to Alexa 488 to reveal airways (green) and anti-mCherry to reveal RSV infected cells (magenta). White dotted squares delineate the region presented in a magnified view in the bottom panel. (**B**) 3D surface rendering of airways (green) and RSV infected cells displayed as spots in a portion of the lung (59.9 mm^3^) at 2 d.p.i. Spots are displayed in different colors according to their distance from airways; magenta when inferior to 4 µm and or white when superior to 4 µm. (**C**) Counting of RSV infected cells displayed as colored spots with the color code corresponding to their distance from airways. (**D**) Quantitation of RSV infected cells based on 3D image analysis.

**Figure 3 viruses-13-00201-f003:**
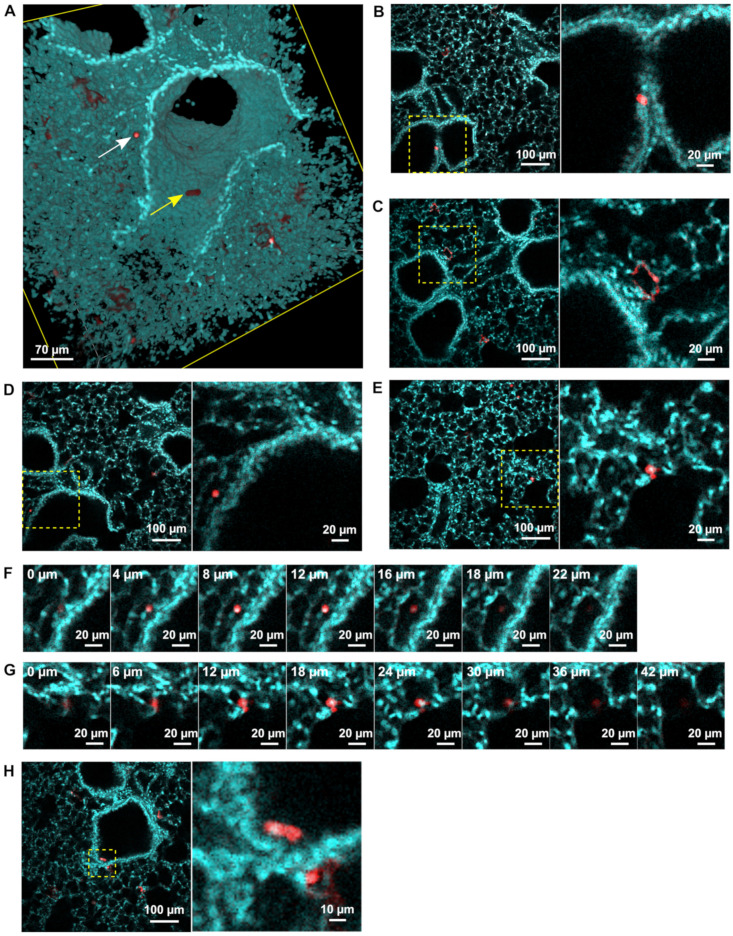
Identification of RSV infected cells in lung biopsies. rHRSV-mCherry infected mouse lung biopsies at 4 d.p.i. were treated with DAPI (cyan) and anti-mCherry antibody (red) to detect cells nuclei and RSV infected cells, respectively. Immunostained biopsies were acquired by 2-photon microscopy post clearing. (**A**) 3D rendering of 3D acquisitions showing a desquamated infected cell in the lumen of the airway (yellow arrow) and a macrophage in the alveolar lumen (white arrow). Optical sections of representative 2-photon acquisitions showing mCherry + epithelial cell (**B**), type I pneumocytes (**C**), alveolar macrophage (**D**) and type II pneumocyte infected by RSV (**E**). Yellow dotted squares delineate the region presented in a magnified view in the bottom panel. Selection of successive optical sections showing an alveolar macrophage (**F**) localized in the alveolar lumen and a type II pneumocyte (**G**) lining the alveoli wall. Depth is indicated in the upper-left corner of each image. Optical section showing a desquamated infected cell in the lumen of the airway (**H**).

**Figure 4 viruses-13-00201-f004:**
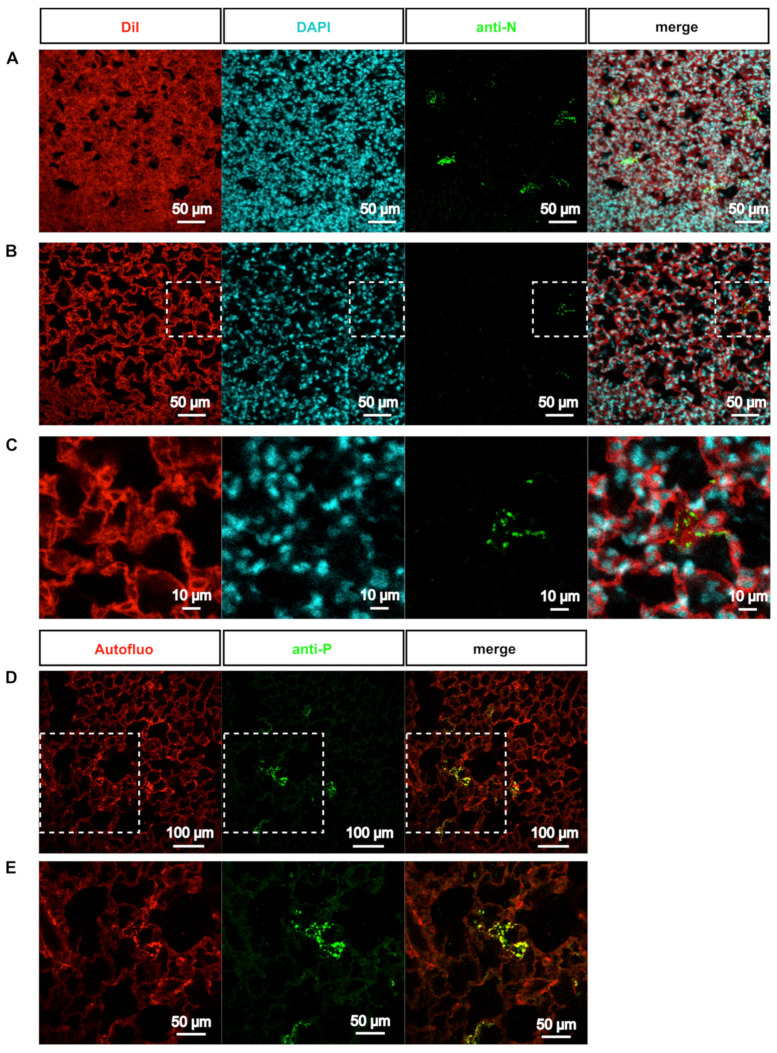
Visualization of RSV inclusion bodies *ex vivo*. (**A**-**C**) Biopsies of rHRSV-mCherry infected mouse lung were stained to visualize cell membrane and cytoplasm (DiI, red), nucleus (DAPI, cyan) and RSV inclusion bodies (anti-N antibody, green). After clearing, immunostained biopsies were acquired by confocal microscopy. (**A**) Maximal projection of a 30 µm z-stack showing inclusion bodies in sparse cells of the parenchyma. (**B**) Representative optical section showing inclusion bodies in the cytoplasm of infected cells. White dotted squares delineate the magnified view presented in (**C**). (**D**) Biopsies of rHRSV-mCherry infected mouse lung were stained with anti-P antibody (green) to visualize RSV inclusion bodies. Autofluorescence (red) was acquired to visualize tissues. Representative optical section. White dotted squares delineate the magnified view presented in (**E**).

## Data Availability

The data presented in this study are available on request from the corresponding author.

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
