# Peer review of "New Look at RSV Infection: Tissue Clearing and 3D Imaging of the Entire Mouse Lung at Cellular Resolution"

_viruses, 2021, doi:10.3390/v13020201_

Round 1

Reviewer 1 Report

Comments

  1. In the materials and methods, currently little to no information is available on the virus used in the assays. What is the source of rHRSV-Luc and rHRSV-mCherry? How produced? What were the titers of the RSV stocks? What was the infectious input in the experiments? In summary, the methods by which they obtained their virus must be explained extensively.
  2. Why bioluminescence signals of rHRSV-Luc were detected in the nasal turbinates and the lungs but not in the trachea? Is RSV unable to infect the trachea, or is the virus titer insufficient?
  3. In Results 3.2, no obvious pathological changes in the lung tissues were observed by 3D reconstruction. Whether 3D reconstruction was not used to observe lung inflammation, the titers of rHRSV were low, or individual differences between mice?
  4. IBs usually included 4 proteins, N, P, L and M2-1. In this manuscript, N protein positive signals were considered as IBs. I do not think the results were rigorous. It is recommended that the author should supply the data of N, P, L and M2-1 co-staining to identify IBs.

Author Response

  1. In the materials and methods, currently little to no information is available on the virus used in the assays. What is the source of rHRSV-Luc and rHRSV-mCherry? How produced? What were the titers of the RSV stocks? What was the infectious input in the experiments? In summary, the methods by which they obtained their virus must be explained extensively.

The methods have already been described in the reference 48 indicated in the manuscript. However, we edited the Materiel and Methods following the reviewer recommendation. P4, lines 107-114. “Recombinant RSV viruses rHRSV-mCherry and rHRSV-Luc corresponding to RSV Long strain expressing either the mCherry or the Luciferase proteins respectively were propagated in HEp-2 cells (ATCC number CCL-23) by infecting a freshly prepared confluent monolayer grown in MEM supplemented with 2% of FCS as previously described [1]. When the cytopathic effect involved the whole monolayer, the viral suspension was collected from the infected cell supernatant and clarified by centrifugation. Titration of the rHRSV-mCherry and rHRSV-Luc virus stocks was determined by plaque assay on HEp-2 cell monolayers at 1.2× 106 pfu/ml and 1.4 × 107 pfu/mL, respectively. “

  1. Why bioluminescence signals of rHRSV-Luc were detected in the nasal turbinates and the lungs but not in the trachea? Is RSV unable to infect the trachea, or is the virus titer insufficient?

Thank you for this comment. To the best of our knowledge, RSV infection in tracheal cells had never been demonstrated in vitro and in vivo. However, we cannot exclude that the lack of detection of the virus in the trachea in our experiments results from an insufficient virus titer. According to the reviewer’s comment, we have moderated our conclusions. P6, lines 196-198: “These results can be related to the dose of the virus administered by intranasal instillation and do not exclude the ability of RSV to infect tracheal cells in other conditions.”

  1. In Results 3.2, no obvious pathological changes in the lung tissues were observed by 3D reconstruction. Whether 3D reconstruction was not used to observe lung inflammation, the titers of rHRSV were low, or individual differences between mice?

According to the reviewer’s comment, P10, lines 250-255 we added the following sentence: “These observations correlate with previous data showing that in this model RSV replication occurred from 2-4 d.p.i. before observation of immune cell infiltration from 5-7 d.p.i. leading to tissue damage [2, 3]. Further studies may be considered to characterize more precisely the activation of resident or recruited immune cells in response to the viral infection, at different time point, and using specific immunolabeling of lymphocytes, leucocytes and/or macrophages.

  1. IBs usually included 4 proteins, N, P, L and M2-1. In this manuscript, N protein positive signals were considered as IBs. I do not think the results were rigorous. It is recommended that the author should supply the data of N, P, L and M2-1 co-staining to identify IBs.

We agree with the reviewer that co-staining of almost 2 proteins of the polymerase complex (in general N and P) should be required to strongly support our data. However, it is well accepted that IBs can be detect based on the observation of cytoplasmic inclusions using anti-P and anti-N alone. As mentioned in the manuscript, anti-P labeling using rabbit polyclonal antibodies allows performing similar observations than N labeling. [4, 5]

To answer the reviewer’s comment, the paragraph was revised and changed by P12, lines 272-276 “Higher magnification of positive signals confirmed the presence of cytosolic fluorescent puncta in type I pneumocytes (Figure 4C). Of note, similar results were obtained when labelling the P protein (data not shown). These observations strongly suggest that cytosolic puncta could correspond to RSV IBs, although a co-staining by both anti-N and anti-P antibodies is nedeed to confirm this hypothesis.

[1] Rameix-Welti MA, Le Goffic R, Herve PL, Sourimant J, Remot A, Riffault S, et al. Visualizing the replication of respiratory syncytial virus in cells and in living mice. Nat Commun. 2014;5:5104.

[2] Lee YT, Kim KH, Hwang HS, Lee Y, Kwon YM, Ko EJ, et al. Innate and adaptive cellular phenotypes contributing to pulmonary disease in mice after respiratory syncytial virus immunization and infection. Virology. 2015;485:36-46.

[3] Roux X, Dubuquoy C, Durand G, Tran-Tolla TL, Castagne N, Bernard J, et al. Sub-nucleocapsid nanoparticles: a nasal vaccine against respiratory syncytial virus. PLoS One. 2008;3:e1766.

[4] Galloux M, Risso-Ballester J, Richard CA, Fix J, Rameix-Welti MA, Eleouet JF. Minimal Elements Required for the Formation of Respiratory Syncytial Virus Cytoplasmic Inclusion Bodies In Vivo and In Vitro. mBio. 2020;11.

[5] Rincheval V, Lelek M, Gault E, Bouillier C, Sitterlin D, Blouquit-Laye S, et al. Functional organization of cytoplasmic inclusion bodies in cells infected by respiratory syncytial virus. Nat Commun. 2017;8:563.

[6] Li G, Fox SE, Summa B, Hu B, Wenk C, Akmatbekov A, et al. Multiscale 3-dimensional pathology findings of COVID-19 diseased lung using high-resolution cleared tissue microscopy. bioRxiv. 2020.

[7] Bryche B, Fretaud M, Saint-Albin Deliot A, Galloux M, Sedano L, Langevin C, et al. Respiratory syncytial virus tropism for olfactory sensory neurons in mice. J Neurochem. 2020;155:137-53.

[8] Eckermann M, Frohn J, Reichardt M, Osterhoff M, Sprung M, Westermeier F, et al. 3D virtual pathohistology of lung tissue from Covid-19 patients based on phase contrast X-ray tomography. Elife. 2020;9.

[9] Matryba P, Kaczmarek L, Gołąb J. Advances in Ex Situ Tissue Optical Clearing. Laser & Photonics Reviews. 2019;13.

[10] Scott GD, Blum ED, Fryer AD, Jacoby DB. Tissue optical clearing, three-dimensional imaging, and computer morphometry in whole mouse lungs and human airways. Am J Respir Cell Mol Biol. 2014;51:43-55.

[11] Tian T, Yang Z, Li X. Tissue clearing technique: Recent progress and biomedical applications. J Anat. 2020.

Reviewer 2 Report

This manuscript describes the location of RSV in the airway in an intact mouse model 2 and 4 days after infection of the airway. The methods are elegant, and the results are interesting, although how they would be applied, especially to human infection, is uncertain. First, mouse disease is different from that of humans. Mice develop pneumonia from RSV infection, while humans have much more bronchiolar involvement. Also, it’s difficult to see how these observations, if completed in humans, expand what we already know about RSV in humans from existing studies. The importance of the manuscript, besides its impressive procedural establishment of these techniques, would seem to be strengthened greatly by discussion of these more clinical matters.

The results are well-presented, although some clarification is necessary:

There are many references to “tissue clearing”, but there is not an explanation of exactly what this is. Does it mean removal of substances that interfere with the visualization assays, or is it a step toward improved analysis of the pathologic changes?

In Figure 1B, does the figure show the total of viral replication in the nose and lung together? The manuscript says there is replication in the nose on day 2, but not in the lung until day 4. The figure does not seem to indicate this. We ask because intranasal inoculation of RSV to an anaesthetized animal results in immediate deposition of virus in both the lung and nose. It would therefore not be expected, necessarily, that lung infection would follow behind nasal infection (although, that’s certainly possible).

Figure 1C. There is not a description of what the different colors indicate. What is red, what is blue? It seems to represent how much virus is present in different locations, but this is not described in the legend.

Line 220. What is meant by “segmented airway epithelium”?

Line 266 states there is no evidence of a strong inflammatory response. In this model, virus replicates well in the airway and lung from day 3-5, but strong cellular infiltration into the lung is not observed until days 5-7. The sentence should add this explanation.

Line 315 describes the manuscript as constituting a “major advance in the understanding of RSV pathogenesis.” That is partially true, but disease in the mouse and in humans seems to be determined as much or more by the inflammatory response as by the degree of viral replication. That is, mice show the greatest degrees of tachypnea after the viral load is declining, and the inflammatory response is increasing. The same seems to be true in humans, in whom administration of antivirals at the time of admission reduces viral replication, but does not alter the course of illness. Therefore, a true advance in the understanding of pathogenesis would seem to require study of the same parameters through later times in the course of infection.

Author Response

  1. This manuscript describes the location of RSV in the airway in an intact mouse model 2 and 4 days after infection of the airway. The methods are elegant, and the results are interesting, although how they would be applied, especially to human infection, is uncertain. First, mouse disease is different from that of humans. Mice develop pneumonia from RSV infection, while humans have much more bronchiolar involvement. Also, it’s difficult to see how these observations, if completed in humans, expand what we already know about RSV in humans from existing studies. The importance of the manuscript, besides its impressive procedural establishment of these techniques, would seem to be strengthened greatly by discussion of these more clinical matters.

We agree with the reviewer that the mouse model failed to completely recapitulate the RSV human disease. However, our data aims to establish a proof of concept for the use of novel imaging methods to assess tissue infection and anatomopathology in 3 dimensions. The established method will be suitable to support basic research on viral infections of lungs from various species [6] This will allow enriching the knowledge acquired through intravital imaging (bioluminescence) [1, 7], medical X-ray imaging [8] and classic 2D histology from micrometric sections of tissue [7]. The use of tissue clearing in human health- and more particularly for respiratory diseases- emerges and has already been reported for the analysis of lung organoids or lung biopsies from human patients [9, 10]. In combination with biochemistry and histology, tissue clearing and 3D imaging will contribute to personalized medicine[11].

The text has been modified accordingly P13, lines 287-296 “Mice are often used as model for RSV infection studies. However, the relevance of the mouse model for the study of RSV infection is still debated, as mice do not completely recapitulate the RSV human disease. The established method will be suitable to support basic research on viral infections of lungs from various species [6] This will allow enriching the knowledge acquired through intravital imaging (bioluminescence) [1, 7], medical X-ray imaging [8] and classic 2D histology from micrometric sections of tissue [7]. The use of tissue clearing in human health- and more particularly for respiratory diseases- emerges and has already been reported for the analysis of lung organoids or lung biopsies from human patients [9, 10]. In combination with biochemistry and histology, tissue clearing and 3D imaging will contribute to personalized medicine[11].

The results are well-presented, although some clarification is necessary:

  1. There are many references to “tissue clearing”, but there is not an explanation of exactly what this is. Does it mean removal of substances that interfere with the visualization assays, or is it a step toward improved analysis of the pathologic changes?

The text has been edited according to reviewer’s recommendations: P3, lines 73-82, “Recent advances in tissue clearing techniques (which aims to render tissue optically transparent through reduction of light scattering and light absorption) have fostered the emergence of deep imaging methods of large samples without the need of physical sectioning. A large numbers of methods have been described grouped in 4 main categories based on the chemistry used for the clearing. They all consist in homogenization of refractive index of the whole tissue achieved by the use of organic solvents, high refractive index aqueous solutions, hyperydrating solutions or hydrogel embedding. Once transparent, the samples: ie biopsies, organoids or organisms can be observed to describe complex biological networks in three dimensions at cellular resolution. Therefore, combination of tissue clearing and deep imaging opened unprecedent opportunities in term of anatomy, diagnosis or host-pathogen interaction studies”

  1. In Figure 1B, does the figure show the total of viral replication in the nose and lung together? The manuscript says there is replication in the nose on day 2, but not in the lung until day 4. The figure does not seem to indicate this. We ask because intranasal inoculation of RSV to an anaesthetized animal results in immediate deposition of virus in both the lung and nose. It would therefore not be expected, necessarily, that lung infection would follow behind nasal infection (although, that’s certainly possible).

We thank the reviewer to point this error. Figure 1B represented the quantification of luciferase activity only in the lungs of mice. This has been corrected now in the legend of figure 1B.

This kinetics of replication in mice supports the results of other studies of our group and other laboratories. Thus we add the following sentence to the manuscript P6, lines 192 “as previously published [1]”.

  1. Figure 1C. There is not a description of what the different colors indicate. What is red, what is blue? It seems to represent how much virus is present in different locations, but this is not described in the legend.

We apologize for this oversight. Colors correspond to the scale bar indicating the average radiance: the sum of the photons per second from each pixel inside the region of interest/number of pixels (p/s/cm2/sr), which is representative of viral replication.

The legend of figure 1 has been revised.

  1. Line 220. What is meant by “segmented airway epithelium”?

Segmentation is a process of image analysis, which is used to isolate an object of interest from the image in order to perform analysis of the object.

We agreed that it may be confusing and removed the term « segmented » from the main text. It is now mentioned in the image analysis section of the Materials and Methods.

  1. Line 266 states there is no evidence of a strong inflammatory response. In this model, virus replicates well in the airway and lung from day 3-5, but strong cellular infiltration into the lung is not observed until days 5-7. The sentence should add this explanation.

We thank the reviewer for this comment. The manuscript has been edited P10, lines 250-255 “These observations correlate with previous data showing that in this model RSV replication occurred from 2-4 d.p.i. before observation of immune cell infiltration from 5-7 d.p.i. leading to tissue damage [2, 3]. Further studies may be considered to characterize more precisely the activation of resident or recruited immune cells in response to the viral infection, at different time point, and using specific immunolabeling of lymphocytes, leucocytes and/or macrophages.

  1. Line 315 describes the manuscript as constituting a “major advance in the understanding of RSV pathogenesis.” That is partially true, but disease in the mouse and in humans seems to be determined as much or more by the inflammatory response as by the degree of viral replication. That is, mice show the greatest degrees of tachypnea after the viral load is declining, and the inflammatory response is increasing. The same seems to be true in humans, in whom administration of antivirals at the time of admission reduces viral replication, but does not alter the course of illness. Therefore, a true advance in the understanding of pathogenesis would seem to require study of the same parameters through later times in the course of infection.

We fully agree with the reviewer. The conclusion has been revised accordingly P12, lines 282-284 “Obtaining and processing of 3D image datasets open new insights into the understanding of RSV infection in complement to 3D transmission electron microscopy, electron tomography and focused ion beam scanning electron microscopy to which it should be compared.”

[1] Rameix-Welti MA, Le Goffic R, Herve PL, Sourimant J, Remot A, Riffault S, et al. Visualizing the replication of respiratory syncytial virus in cells and in living mice. Nat Commun. 2014;5:5104.

[2] Lee YT, Kim KH, Hwang HS, Lee Y, Kwon YM, Ko EJ, et al. Innate and adaptive cellular phenotypes contributing to pulmonary disease in mice after respiratory syncytial virus immunization and infection. Virology. 2015;485:36-46.

[3] Roux X, Dubuquoy C, Durand G, Tran-Tolla TL, Castagne N, Bernard J, et al. Sub-nucleocapsid nanoparticles: a nasal vaccine against respiratory syncytial virus. PLoS One. 2008;3:e1766.

[4] Galloux M, Risso-Ballester J, Richard CA, Fix J, Rameix-Welti MA, Eleouet JF. Minimal Elements Required for the Formation of Respiratory Syncytial Virus Cytoplasmic Inclusion Bodies In Vivo and In Vitro. mBio. 2020;11.

[5] Rincheval V, Lelek M, Gault E, Bouillier C, Sitterlin D, Blouquit-Laye S, et al. Functional organization of cytoplasmic inclusion bodies in cells infected by respiratory syncytial virus. Nat Commun. 2017;8:563.

[6] Li G, Fox SE, Summa B, Hu B, Wenk C, Akmatbekov A, et al. Multiscale 3-dimensional pathology findings of COVID-19 diseased lung using high-resolution cleared tissue microscopy. bioRxiv. 2020.

[7] Bryche B, Fretaud M, Saint-Albin Deliot A, Galloux M, Sedano L, Langevin C, et al. Respiratory syncytial virus tropism for olfactory sensory neurons in mice. J Neurochem. 2020;155:137-53.

[8] Eckermann M, Frohn J, Reichardt M, Osterhoff M, Sprung M, Westermeier F, et al. 3D virtual pathohistology of lung tissue from Covid-19 patients based on phase contrast X-ray tomography. Elife. 2020;9.

[9] Matryba P, Kaczmarek L, Gołąb J. Advances in Ex Situ Tissue Optical Clearing. Laser & Photonics Reviews. 2019;13.

[10] Scott GD, Blum ED, Fryer AD, Jacoby DB. Tissue optical clearing, three-dimensional imaging, and computer morphometry in whole mouse lungs and human airways. Am J Respir Cell Mol Biol. 2014;51:43-55.

[11] Tian T, Yang Z, Li X. Tissue clearing technique: Recent progress and biomedical applications. J Anat. 2020.